# Differentiable Architecture Search for Reinforcement Learning

**Yingjie Miao**[*], **Xingyou Song**[*], **John D. Co-Reyes, Daiyi Peng, Summer Yue, Eugene Brevdo, Aleksandra Faust**

Google Research, Brain Team

**Abstract**    In this paper, we investigate the fundamental question: *To what extent are gradient-based neural architecture search (NAS) techniques applicable to RL?* Using the original DARTS as a convenient baseline, we discover that the discrete architectures found can achieve up to 250% performance compared to manual architecture designs on both discrete and continuous action space environments across off-policy and on-policy RL algorithms, at only 3x more computation time. Furthermore, through numerous ablation studies, we systematically verify that not only does DARTS correctly upweight operations during its supernet phrase, but also gradually improves resulting discrete cells up to 30x more efficiently than random search, suggesting DARTS is surprisingly an effective tool for improving architectures in RL.

## 1 Introduction and Motivation

Over the past few years, Differentiable Architecture Search (DARTS) (Liu et al., 2019) has dramatically risen in popularity as a method for neural architecture search (NAS), with multiple modifications and improvements (Chu et al., 2020c; Li et al., 2021; Chu et al., 2020a; Liang et al., 2019; Wang et al., 2021a; Chu et al., 2020b; Hundt et al., 2019; Wang et al., 2021b; Chen and Hsieh, 2020; Zela et al., 2020) constantly proposed at a rapid speed. From a broader perspective, one may wonder how far we may push the limits of differentiable search, as differentiability is the cornerstone of all deep learning

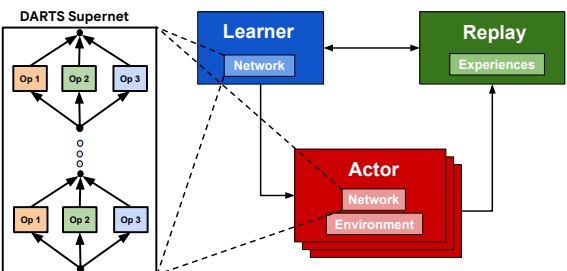

**Figure 1**: DARTS is potentially a natural fit for RL, as a DARTS supernet can simply be inserted into the network components of a standard RL training pipeline, which may potentially be highly distributed.

research. One such very large application space is in reinforcement learning (RL), with DARTS being a natural fit due to its simplicity and modularity in integrating with large distributed RL training pipelines. Shockingly however, among the vast amounts of literature on DARTS, there are virtually no works addressing its viability in RL.

This can be attributed to the fact that RL fundamentally does not follow the same optimization paradigm as supervised learning (SL). The end goal of RL is not to simply minimize the loss or accuracy over a fixed dataset, but rather to improve a policy's reward over an entire environment whose training data is generated by the policy itself. This scenario raises the possibility of a negative feedback loop in which a poorly trained policy may achieve trivial reward even if it successfully optimizes its loss to zero, and thus the loss is not an indicator of a policy's true performance. As DARTS is purely reliant on optimizing the architecture topology by using only the loss as the search signal, therein lies the simple question: *Would DARTS even work in RL?*

---

[*]Equal contribution. Order decided randomly.

Correspondence to: {yingjiemiao,xingyousong,sandrafaust}@google.com

Code can be found at https://github.com/google/brain_autorl/tree/main/rl_darts.

It is highly valuable to answer this question, as recently, works in larger-scale RL suggest that policy architecture designs greatly affect various metrics such as generalization, transferrability, and efficiency. One surprising phenomenon found on Procgen (Cobbe et al., 2020), a procedural generation benchmark for RL, was that "IMPALA-CNN", a residual convolutional architecture from (Espeholt et al., 2018), could substantially outperform "NatureCNN", the standard 3-layer architecture used for Atari (Mnih et al., 2013), in both generalization and sample complexity under limited and infinite data regimes respectively (Cobbe et al., 2019). Furthermore, in robotics subfields such as grasping (Kalashnikov et al., 2018; Rao et al., 2020), cameras collect very detailed real-world images (ex: 472 x 472, 4x larger than ImageNet (Russakovsky et al., 2015)) for observations which require deep image encoders consisting of more than 15 convolutions, raising concerns on efficiency and speed in policy training and inference. As such RL policy networks gradually become larger and more sophisticated, so does the need for understanding and automating such designs.

In this paper, we show that DARTS can in fact find better architectures efficiently in RL, one of the first instances in which the loss does not directly affect the final objective. This work may be of interest to both the gradient-based NAS community as means to expand to the RL domain, and the AutoRL community as a practical toolset and method for co-training policies and neural architectures for better performing agents. Our contributions are:

- We conceptually identify the key differences between SL and RL in terms of their usage of the loss function, which raise important hypothetical questions and issues about whether DARTS is applicable to RL. In particular, these deal with the quality of the training signal to the architecture variables during supernet training, and downstream effects on discrete cells during evaluation.

- Empirically, we find that DARTS is in fact compatible with several on-policy and off-policy algorithms including PPO (Schulman et al., 2017), Rainbow-DQN (Hessel et al., 2018), and SAC (Haarnoja et al., 2018). The discrete architectures found can reach up to 250% performance compared to manual architecture designs on discrete action (e.g. Procgen) and continuous control (e.g. DM-Control) environments, at only 3x more computation time.

- Through comprehensive ablation studies, we show the supernet successfully trains, and reasonably upweights optimal operations. We further verify both qualitatively and quantitatively that discretized cells gradually evolve to better architectures. However, we also demonstrate how this can fail, especially if the corresponding supernet fails to train, with further extensive ablations in the Appendix.

**Related Works**. Recently, there have been a flurry of works modifying many components in the RL pipeline, both manually and automatically, as part of the broader Automated Reinforcement Learning (AutoRL) (Parker-Holder et al., 2022) field. Specifically for architecture components, manual modifications include (Raileanu and Fergus, 2021; Nafi et al., 2021; Tang and Ha, 2021) which have shown great success in improving metrics such as sample complexity and generalization, especially on the Procgen benchmark. However, very few works have considered the possibility of actually *automating* the search for new architectures, i.e. "NAS for RL", specifically for large-scale modern convolutional networks.

Most previous NAS for RL works only involve small policies trained via blackbox/evolutionary optimization methods, which include (Song et al., 2021; Gaier and Ha, 2019; Stanley and Miikkulainen, 2002; Stanley et al., 2009), utilizing CPU workers for forward pass evaluations rather than exact gradient computation on GPUs. Such methods are usually unable to train policies involving more than 10K+ parameters due to the sample complexity of zeroth order methods in high dimensional parameter space (Agarwal et al., 2010). The only previous known application of gradient-based routing is (Akinola et al., 2021), which searches for the optimal way of combining observation and action tensors together in off-policy QT-OPT (Kalashnikov et al., 2018), but does

---
**Algorithm 1**: RL-DARTS Procedure.
---
  1. **Supernet training**: Compute $\alpha^*$ from $\arg\max_{\theta,\alpha} J(\pi_{\theta,\alpha})$ via $\arg\min_{\theta,\alpha} \mathcal{L}^{RL}(\theta,\alpha)$.
  2. **Discretization**: Discretize $\alpha^*$ to construct evaluation policy $\pi_{\phi,\delta(\alpha^*)}$.
  3. **Evaluation**: Report $\max_\phi J(\pi_{\phi,\delta(\alpha^*)})$ via $\arg\min_\phi \mathcal{L}_{\delta(\alpha^*)}(\phi)$.
---

not search for image encoders nor uses the supernet for inference, as it trains using off-policy robotic data collected independently. This leaves the applicability of DARTS to inference-dependent RL as an open question addressed in our work.

## 2 Problem Overview and Method

**DARTS Preliminaries**. Since we only use the original DARTS (Liu et al., 2019) to reduce confounding factors, we thus give a very brief overview of DARTS to save space. More comprehensive details can be found in Appendix G.5 and the original paper. DARTS optimizes substructures called cells, where each cell contains $I$ intermediate nodes organized in a directed acyclic graph, where each node $x^{(i)}$, represents a feature map, and each edge $(i,j)$ consists of an operation (op) $o^{(i,j)}$, with later nodes $x^{(j)}$ merged (e.g. summation) from some previous $o^{(i,j)}(x^{(i)})$. A DARTS supernet is constructed by continuously relaxing selection of ops in $\mathcal{O}$, via softmax weighting, i.e. $\overline{o}^{(i,j)}(x^{(i)}) = \sum_{o \in \mathcal{O}} p_o^{(i,j)} \cdot o(x^{(i)})$, where $p_o^{(i,j)} = \frac{\exp(\alpha_o^{(i,j)})}{\sum_{o' \in \mathcal{O}} \exp(\alpha_{o'}^{(i,j)})}$. The cell's output is by default the result of a Conv1x1 op on the depthwise concatenation of all intermediate node features, although this may be changed (e.g. by simply outputting the last intermediate node's features). We denote the collection of all architecture variables $a_o^{(i,j)}$ as $\alpha$. Denote the total set of possible operations in our searchable network as $\mathcal{O} = \mathcal{O}_{base} \cup \{\text{Zero, Skip}\}$ which must contain Zero and Skip Connection ops, while $\mathcal{O}_{base}$ is user-defined. Denote $\alpha$ to be the pre-softmax trainable architecture variables in the supernet. A predefined loss function $\mathcal{L}(\theta)$ over neural network weights $\theta$ will thus be redefined as $\mathcal{L}(\theta,\alpha)$ when under DARTS's *search mode*, where the original model $f_\theta$ will be replaced with a dense supernet $f_{\theta,\alpha}$. During evaluation time, a trained $\alpha^*$ is then *discretized* into a sparser final cell $\delta(\alpha^*)$ by representing each edge $(i,j)$ with the highest softmax weighted op, i.e. $\arg\max_{o \in \mathcal{O}, o \neq zero} p_o^{(i,j)}$, and then retaining only the top $K$ incoming edges for each intermediate node. We thus retrain over the new loss $\mathcal{L}_{\delta(\alpha^*)}(\phi)$, now dependent on only fresh sparse weights $\phi$ to obtain the final reported metric.

**RL Preliminaries**. For RL notation, given an MDP $\mathcal{M}$, denote $s_t, a_t, r_t$ as state, action, reward respectively at time $t$. $\pi$ is the policy and $\mathcal{D}$ is the replay buffer containing collected trajectories $\tau = (s_0, a_0, r_0, s_1, \ldots)$. The goal is to maximize $J(\pi) = \mathbb{E}_{\tau \sim \pi}\left[\sum_{t \geq 0} r_t\right]$, the expected cumulative reward using policy $\pi$. In most RL algorithms, there is the notion of a neural network torso or *encoder* $f_\theta$ mapping the state $s$ to a final feature vector when forming $\pi$. In the DARTS case, we use a supernet encoder $f_{\theta,\alpha}$ leading to a supernet policy denoted as $\pi_{\theta,\alpha}$ and also a corresponding discretized-cell policy $\pi_{\phi,\delta(\alpha^*)}$ for evaluation.

### 2.1 Methodology

SL features the notion of training and validation sets, with corresponding losses $\mathcal{L}_{train}^{SL}, \mathcal{L}_{val}^{SL}$, where the learning procedure consists of a bilevel optimization problem and the goal is to find $\alpha^* = \arg\min_\alpha \mathcal{L}_{val}^{SL}(\theta^*, \alpha)$ where $\theta^* = \arg\min_\theta \mathcal{L}_{train}^{SL}(\theta, \alpha)$. In this paper, we do not need to use the original bilevel optimization framework, as we are optimizing sample complexity and raw training performance, which are standard metrics in RL. Furthermore, bilevel optimization is notoriously difficult and unstable, sometimes requiring special techniques (Liu et al., 2018; Dong and Yang, 2019; Hundt et al., 2019; Li et al., 2020; Chu et al., 2020b,a,c; Liang et al., 2019; Wang et al., 2021b) specific to SL optimization, which can confound the results and message of our paper.

We thus joint optimize both $\theta$ and $\alpha$, and the full procedure is concisely summarized in Algorithm 1 as "RL-DARTS". However, this is easier said than done, as we explain the core issues of applying DARTS to RL below.

**SL vs RL DARTS.** Fundamentally, SL relies on a fixed dataset $\mathcal{D}^{SL} = \{(x_i, y_i) \mid i \geq 1\}$, in which the loss is defined as $\mathcal{L}^{SL}(\theta) = \mathbb{E}_{(x,y)\sim\mathcal{D}}[\ell(f_\theta(x), y)]$ where $\ell(\cdot)$ is defined as mean squared error, cross-entropy loss, or negative log-likelihood depending on application. These losses are strongly correlated or even equivalent to the final objective (e.g. accuracy or density estimation), and this reason can be considered a significantly contributing factor to the success of DARTS in SL. Unfortunately in RL, there are no such guarantees that minimizing the loss $\mathcal{L}^{RL}(\cdot)$ necessarily improves the true objective $J(\cdot)$, for two primary reasons:

1. The RL agent's dataset (a.k.a. replay buffer) $\mathcal{D}^{RL} = \{\tau_i \mid i \geq 1\}$ is significantly non-stationary and self-dependent, as it constantly changes based on the current performance of data collection actors, which themselves are functions of $\theta$. Thus, a negative feedback loop may arise, where $\theta$ produces an actor which collects poor training data, leading to convergence towards an even poorer $\theta'$. While the loss $\mathcal{L}^{RL}(\cdot)$ converges to 0 over low quality data, the reward $J(\cdot)$ still does not increase. This issue is commonplace in RL, such as in any environments which require exploration. Hypothetically in the DARTS case, $\alpha$ can potentially produce the same negative feedback loop by converging to subpar operations and impairing supernet training, also leading to a poor discrete cell $\delta(\alpha)$.

2. The losses are considerably more complex and utilize multiple auxiliary losses which assist training but are never used during evaluation. For PPO, the loss is defined as $\mathcal{L}^{RL}_{PPO}(\theta) = \mathbb{E}_{\tau\sim\mathcal{D}_{RL}}[L_{CLIP}(\theta) - L_{VF}(\theta) + \mathcal{H}(\theta)]$. However, the value function (i.e. $L_{VF}$) nor the policy entropy (i.e. $\mathcal{H}$) are ever used to evaluate final reward. This is similarly the case for off-policy algorithms such as SAC which strongly emphasizes maximizing the entropy $\mathcal{H}$, and Rainbow-DQN which also uses value functions to assist training. The question is thus raised in the DARTS case as to whether such auxiliary losses are actually useful signals, or instead inhibit proper training of $\alpha$, whose main goal is to only maximize $J(\cdot)$ using discrete cell $\delta(\alpha)$.

The core theme of our experiments will be understanding whether these are obstacles to DARTS's application to RL.

# 3 Experiments

**Experiment Setup**: To verify that every component of RL-DARTS works as intended, we seek to answer all questions below, by first presenting end-to-end results, and then further key ablation studies:

1. **End-to-End Performance**: Overall, how do the final discrete cells perform at evaluation, and what gains can we obtain from architecture search? Furthermore, how does RL-DARTS compare against random search?

2. **Supernet Training**: During supernet training, how does the $\alpha$ change? Does $\alpha$ converge towards a sparse solution and select good operations over supernet training?

3. **Discrete Cells**: Even if the supernet trains, do the corresponding discrete cells also improve in evaluation performance throughout $\alpha$'s training? What kinds of failure modes occur?

Following common NAS practices (Zoph et al., 2018; Pham et al., 2018; Zoph and Le, 2016), we construct our supernet (with $I$ intermediate cells) by stacking both normal ($N$ times) and reduction cells ($R$ times) together into *blocks*, which are themselves also stacked together $D$ times (see Figure 2). Reduction cells apply a stride of 2 on the input. Each block possesses its own convolutional

channel depth, used throughout all cells in the block. During search, we train a smaller supernet (i.e. depth 16) to reduce computation time, but evaluate final discretized cells on larger models (i.e. depth 64), with $D = 3$ layers for cheap large-scale runs and $D = 5$ layers for fine-grained A/B testing.

For the operation search space, we consider the following base ops $\mathcal{O}_{base}$ and corresponding algorithms:

- **Classic on PPO**: $\mathcal{O}_{base,N}$ = {Conv3x3+ReLU, Conv5x5+ReLU, Dilated3x3+ReLU, Dilated5x5+ReLU} for normal ops and $\mathcal{O}_{base,R}$ = {Conv3x3, MaxPool3x3, AveragePool3x3} for reduction ops, which is standard in supervised learning (Liu et al., 2019; Zoph et al., 2018; Pham et al., 2018).

- **Micro on Rainbow and SAC**: We also propose $\mathcal{O}_{base,N}$ = {Conv3x3, ReLU, Tanh}, a more fine-grained and novel search space which has not been used previously in SL. The inclusion of Tanh is motivated by its use previously for continuous control architectures (Salimans et al., 2017; Song et al., 2020).

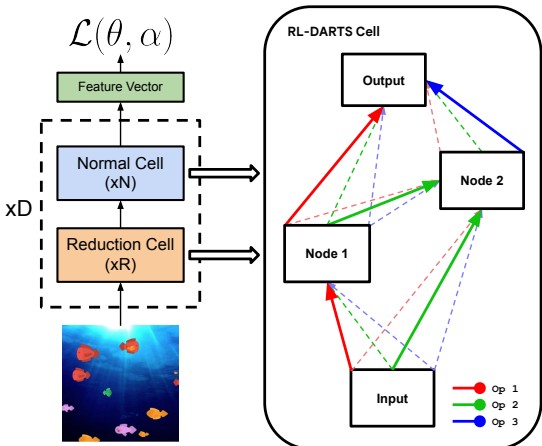

**Figure 2:** Illustration of our network via stacking normal and reduction cells. Solid lines correspond to selected ops after discretization from all possible ops weighted using $\alpha$. If $R > 0$, we add an initial Conv3x3 for preprocessing.

For benchmarks, we primarily use Procgen (Cobbe et al., 2019, 2020) for discrete action spaces, with PPO (Schulman et al., 2017) and Rainbow DQN (Hessel et al., 2018) as training algorithms, with Procgen's difficulty set to "easy" similar to other works (Raileanu et al., 2020; Raileanu and Fergus, 2021; Parker-Holder et al., 2021a). Procgen comprehensively evaluates all aspects of RL-DARTS, as it possesses a diverse selection of 16 games, each with infinite levels to simulate large data regimes where episodes may drastically change, relevant for generalization. It further uses the IMPALA-CNN architecture (Espeholt et al., 2018) as a strong hand-designed baseline, and can be seen as a specific instance of the stacked cell design in Figure 2, where its "Reduction Cell" consists of a Conv3x3 and MaxPool3x3 (Stride 2) with $R = 1$ and its "Normal Cell" consists of a residual layer with Conv3x3's and ReLU's, with $N = 2$. For fair comparisons to IMPALA-CNN, we use $(N, R, I) = (1, 1, 4)$ on the classic search space, while $(N, R, I) = (2, 0, 4)$ on the Micro search space, where reduction ops default to IMPALA-CNN's in order to avoid hidden confounding effects when visualizing Micro cells.

In addition, we also assess DARTS's viability in continuous control which is common in robotics tasks. We use the common DM-Control benchmark (Tassa et al., 2018) along with the popular SAC algorithm. We use $N = 3, I = 4, K = 1$ with "Micro" search space to remain fair to the 4-layer convolutional encoder baseline observing images of size $64 \times 64$ (full details in Appendix G).

Unless specified, we by default use consistent hyperparameters (found in Appendix G) for all comparisons found inside a figure, although learning rate and minibatch size may be altered when training models of different sizes due to GPU memory limits. Thus, even though RL is commonly sensitive to hyperparameters (Zhang et al., 2021), we surprisingly find that **once a pre-existing RL baseline has already been setup, incorporating DARTS requires no extra cost in tuning, as evidence of its ease-of-use**.

### 3.1 End-to-End Results on Multi-task, Discrete and Continuous Control Tasks

**Multi-game Search.** We first begin with the most surprising and largest end-to-end result in terms of scale of data and compute shown in Figure 3: **By training a supernet across infinite levels**

across *all 16 Procgen games* to find a single transferrable cell, we are able to achieve up to **250**% **evaluation performance compared to the IMPALA-CNN baseline** over select environments. This is achieved using Rainbow with our proposed "Micro" search space, where a learner performs gradient updates over actor replay data (with normalized rewards) from all games.

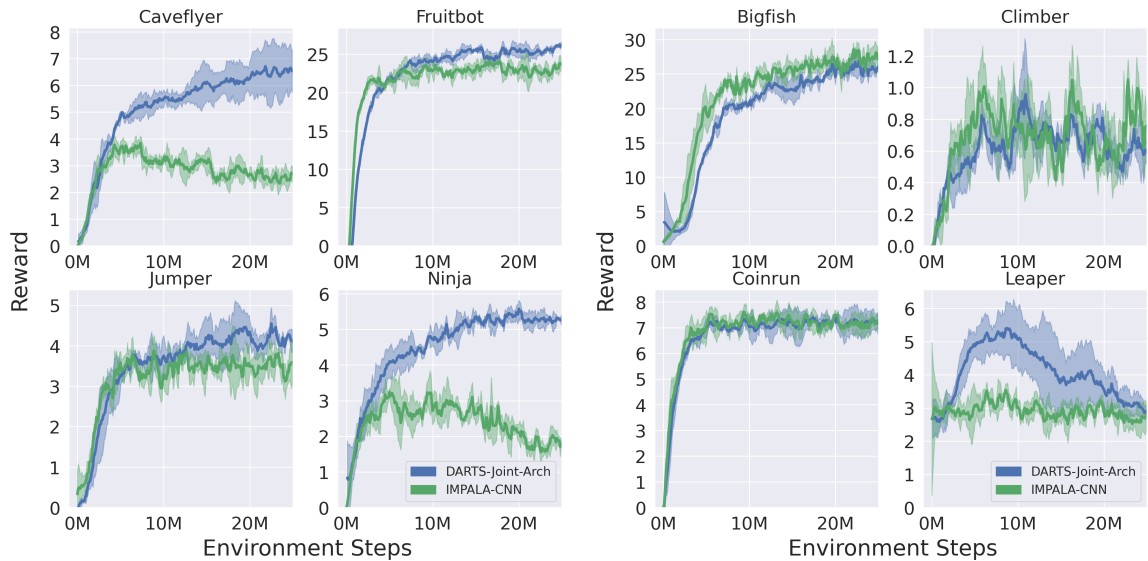

**Figure 3**: Evaluation of the discrete cell joint-trained over 8 environments using depths $64 \times 5$ to emphasize comparison differences.

We further display the discovered discrete cell and the supernet joint-training procedure in Figure 4. Interestingly, the discrete cell uses nonlinearities over all intermediate connections, with convolutions only used via the merge operation for the output.

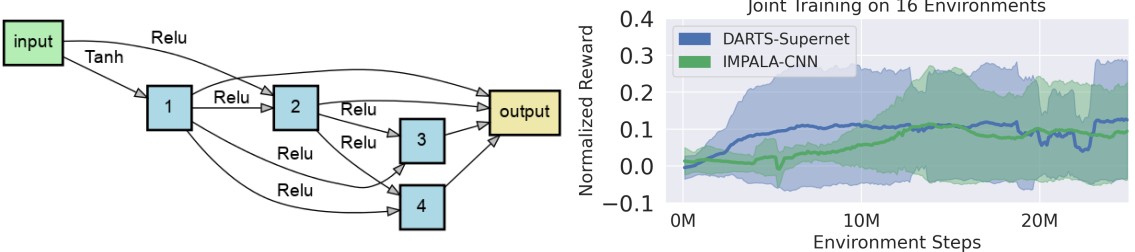

**Figure 4**: **Left**: Discrete cell found. **Right**: Average normalized rewards over all 16 games during supernet + baseline training.

**Single-game Search**. We further compare DARTS's end-to-end performance against random search, but more appropriately applied to single game scenarios. On the PPO side, we use the "Classic" search space (total size $4 \times 10^{11}$, see Appendix H.1). For a fair comparison, we ensure total wall-clock time (with same hardware) stays equal, as common in (Liu et al., 2019). Since in Appendix A, Table 4, a PPO supernet takes 2.5x longer to reach 25M steps, this is rounded to a random search budget of 3 cells to be trained with depths $16 \times 3$ for 25M steps. The best of the 3 cells is used for full evaluation. In Table 1, on average, random search underperforms significantly.

|  | IMPALA-CNN | RL-DARTS (Discrete Cell) | Random Search |
|---|---|---|---|
| Avg. Normalized Reward | 0.708 | 0.709 | 0.489 |

**Table 1**: Average normalized rewards across all 16 environments w/ PPO, using the normalization method from (Cobbe et al., 2020). Full details and results (including Rainbow) are presented in Appendix E.

In Figure 5 we further find that RL-DARTS is capable of finding game-specific architectures from scratch which outperform IMPALA-CNN on select environments such as Plunder and Heist, while maintaining competitive performance on others.

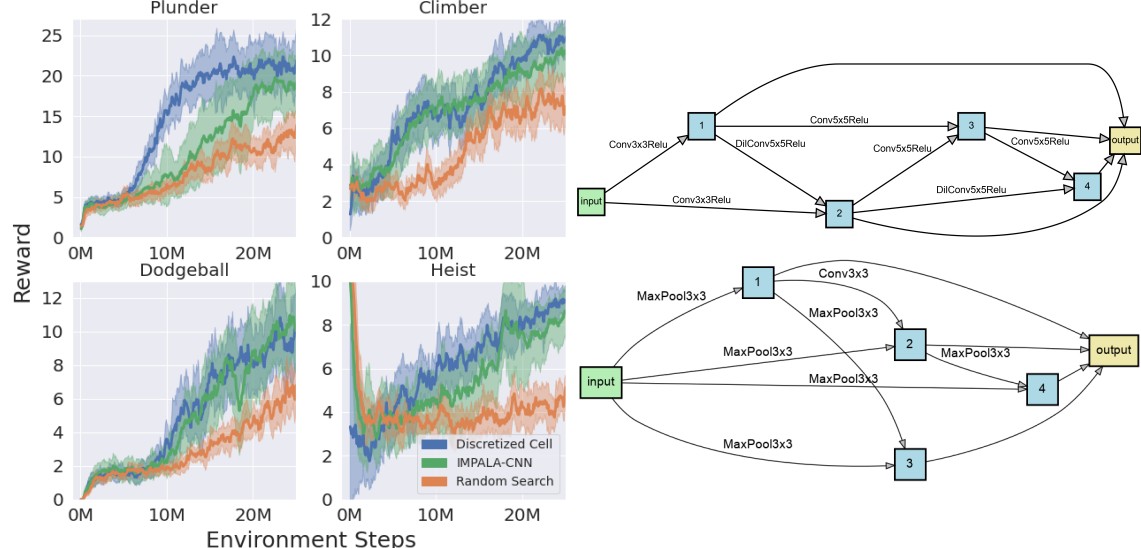

**Figure 5**: **Left 2x2 Plot**: Examples of discrete cell evaluations using the "Classic" search space with PPO, with depths $64 \times 3$. **Right**: Normal (Top) and Reduction (Bottom) cells found for "Plunder" which achieves faster training than IMPALA-CNN. Note the interesting use of 5x5 convolutional kernel sizes later in the cell.

**100 Random Cell Comparison**. To further present a more comprehensive comparison against random search, and understand how strong IMPALA-CNN is as a baseline, we compare against 100 unique random cells. To manage the computational load, we performed the study over two selected environments, as shown in Figure 6 and find that **all of the random cells underperform against both IMPALA-CNN and DARTS**. This demonstrates that DARTS possesses a strong search capability, achieving $100/3 \approx 30x$ efficiency over random search and can discover architectures that match in complexity and performance of highly-tuned, expert designed architectures such as IMPALA-CNN.

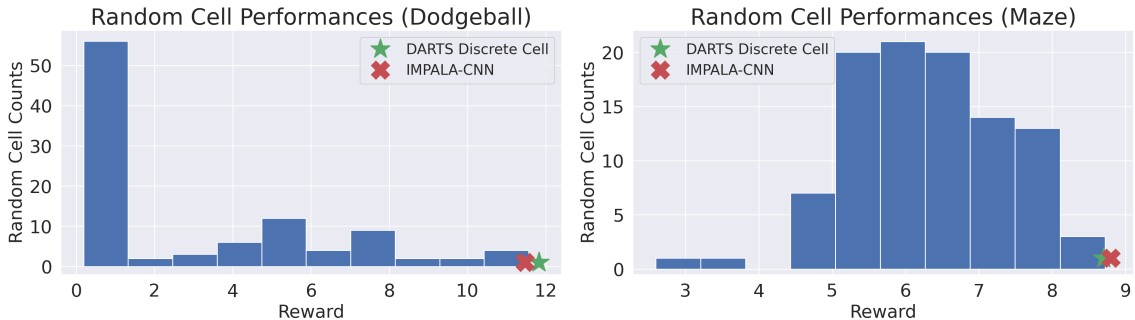

**Figure 6**: Histogram of 100 random cells' rewards over environments Dodgeball and Maze using the Rainbow + "Micro" search space (depths $64 \times 3$), with a significant number of random cells (e.g. 95% for Dodgeball) performing substantially worse than DARTS or IMPALA-CNN.

**DM-Control with Soft Actor-Critic**. DARTS also consistently finds better and stable architectures over multiple lightweight environments involving continuous control (see Figure 7) trained up to 1M steps. Even though the 4-layer encoder network used (details in Appendix G) is significantly smaller than IMPALA-CNN, we find that there is still room for architectural improvement.

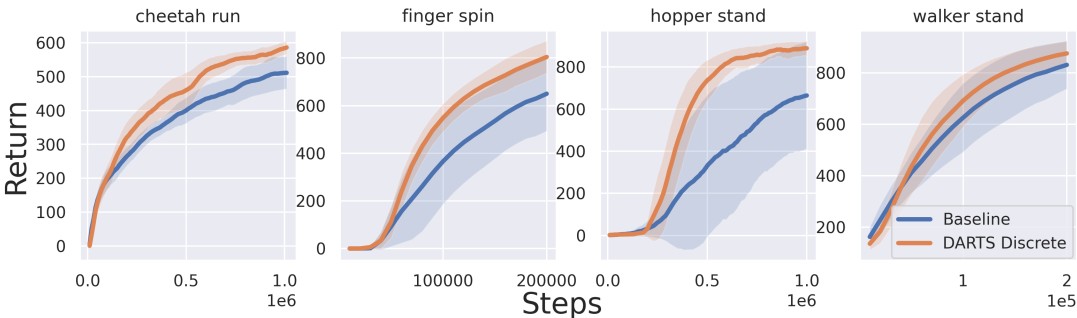

**Figure 7**: RL-DARTS on DM-Control with SAC also finds better architectures over the corresponding baseline.

## 3.2 Role of Supernet Training

In Figure 8, we verify that **training the supernet end-to-end works effectively**, **even with minimal hyperparameter tuning**. Furthermore, the training only requires at 3x more compute time, with extensive efficiency metrics calculated in Appendix A. However, as mentioned in Subsec. 2.1, it is unclear whether the right signals are provided to operation routing variables $\alpha$ via the RL training loss, and whether $\alpha$ produces the correct behavior, which we investigate.

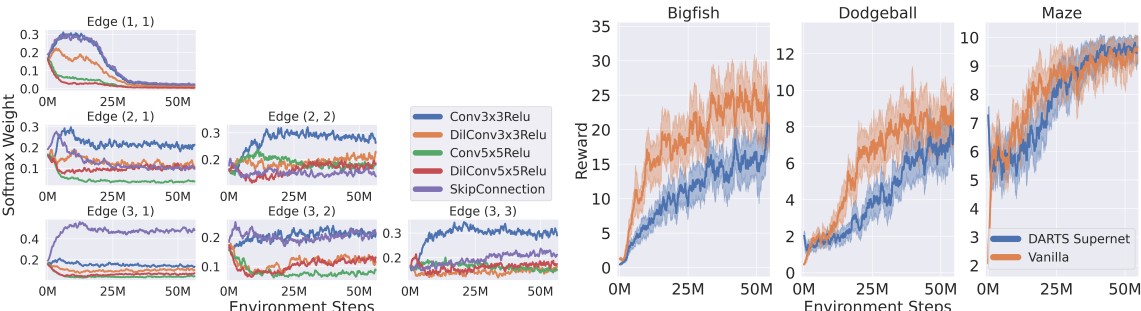

**Figure 8**: **Left**: Softmax op weights over all edges in the cell when training the supernet with PPO + "Classic" search space on Dodgeball. Zero op weight is not shown to improve clarity. **Right**: Sanity check to verify that the supernet eventually achieves a regular training curve, using vanilla IMPALA-CNN as a rough gauge. Both use depths $16 \times 3$. Note that while we show the curve up to 50M steps, we by default discretize $\alpha$ at 25M steps, as op choices have already converged towards a sparser solution. Analogous figure for Rainbow in Appendix B.

Conveniently in Figure 8, we find that $\alpha$ strongly downweights all base ops (in particular, 5x5 ops) except for the standard Conv3x3+ReLU. This provides an opportunity to understand whether $\alpha$ **downweights suboptimal ops throughout training**. We confirm this result in Table 2 by evaluating standard IMPALA-CNN cells using either purely 3x3 or 5x5 convolutions for the whole network, and demonstrating that the 3x3 setting outperforms the 5x5 setting (especially in limited data, e.g. 200-level training/test regime), suggesting the signaling ability of $\alpha$ on op choice.

Answering the converse question is just as important: *Can any supernet train?* The supernet possesses an incredibly dense set of weights, and thus one might wonder whether trainability occurs with any search space or settings. We answer in the negative, where a **poorly designed supernet can fail**. To show this clearly, we remove all ReLU nonlinearities from the "Classic" search space ops used for PPO, as well as simply freeze $\alpha$ to be uniform for Rainbow, and find

| Scenario | Conv 3x3 | Conv 5x5 |
|---|---|---|
| Train (Inf. levels) | **15.1 ± 2.5** | 13.2 ± 2.3 |
| Train (200 levels) | **12.1 ± 1.7** | 9.8 ± 2.1 |
| Test (from Train) | **10.2 ± 2.3** | 5.9 ± 1.7 |

**Table 2**: PPO IMPALA-CNN evaluations (mean return at 50M steps) on Dodgeball. Learning curves can be found in Appendix D.1, Figure 15.

both cases produce poor training in Table 3. **Thus, the supernet in RL provides important search signals in terms of reward and $\alpha$ during training, especially on the quality of a search space.**

|  | Rainbow (25M Steps) | | PPO (50M Steps) | |
|---|---|---|---|---|
| Scenario | Trainable $\alpha$ | Uniform $\alpha$ | With ReLU | No ReLU |
| Training (Inf. levels) Reward | $\mathbf{3.1 \pm 0.5}$ | $0.9 \pm 0.2$ | $\mathbf{7 \pm 0.9}$ | $1.9 \pm 0.3$ |

Table 3: Supernet training rewards on Dodgeball. Learning curves can be found in Appendix C.

### 3.3 Discrete Cell Improvement

We further must analyze whether discrete cells also improve, as they are used for final evaluation and deployment. Strong supernet performance (via continuous relaxation) does not necessarily imply strong evaluation cell performance (via discretization), due to the existence of *integrality gaps* (Wang et al., 2021b,a). Nevertheless, we demonstrate that even using the default $\delta$ discretization from (Liu et al., 2019) leads to both quantitative and qualitative improvements.

**Quantitative improvement**. As $\alpha$ changes during supernet training, so do the outputs of the discretization procedure on $\alpha$. We collect all distinct discrete cells $\{\delta(\alpha_1), \delta(\alpha_2), \ldots\}$ into a sequence, and evaluate each cell's performance $\max_\phi J(\pi_{\phi, \delta(a_i)})$ $\forall i$ via training from scratch for 25M steps (Figure 9). The performance generally improves over time, indicating that supernet optimization selects better cells. However, we find that such behavior can be environment-dependent, as some environments possess less monotonic evaluation curves (see Fig. 18 in Appendix D).

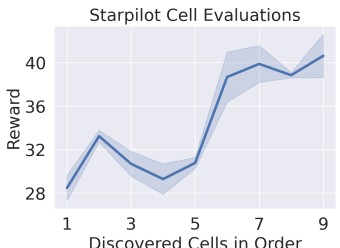

Figure 9: Evaluation of 9 distinct discrete cells in order from the trajectory of $\alpha$ on the Starpilot environment when using Rainbow.

**Qualitative improvement**. We visualize discrete cells $\delta(\alpha_{start}), \delta(\alpha_{end})$ from the start and end of supernet training with Rainbow on the Starpilot environment in Fig. 10. The earlier cell consists of only linear operators is clearly a poor design in comparison to the later cell. We find similar cell evolution results for PPO in Figures 16, 17 in Appendix D.2, displaying more sophisticated yet still interpretable changes in cell topology.

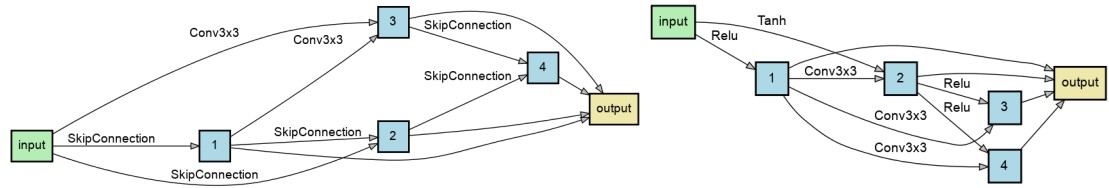

Figure 10: Evolution of discovered cells over a DARTS optimization process. **Left**: $\delta(\alpha_{start})$ discovered in the early stage which is dominated by skip connections and only linear ops. **Right**: $\delta(\alpha_{end})$ discovered in the end which possesses several reasonable local structures similar to Conv + ReLU residual connections.

We further answer one of the most common questions in DARTS research: *What is the direct relationship between a supernet and its discrete cell?* In Figure 11, we provide two supernet runs along with their corresponding discrete cells, side by side in order to answer this question. While "Supernet 1" is a standard successful training run, "Supernet 2" is a failed run which can commonly occur due to the inherent sensitivity and variance in RL training. As it turns out, this directly leads to differences between their corresponding discrete cells both quantitatively and qualitatively as well, in which "Discretized Cell 1"'s design appears to make sense and train properly, while "Discretized Cell 2" is clearly a suboptimal design, and fails to train at all. Thus, a major reason for why a **discretized cell may underperform is if its corresponding supernet fails to learn**.

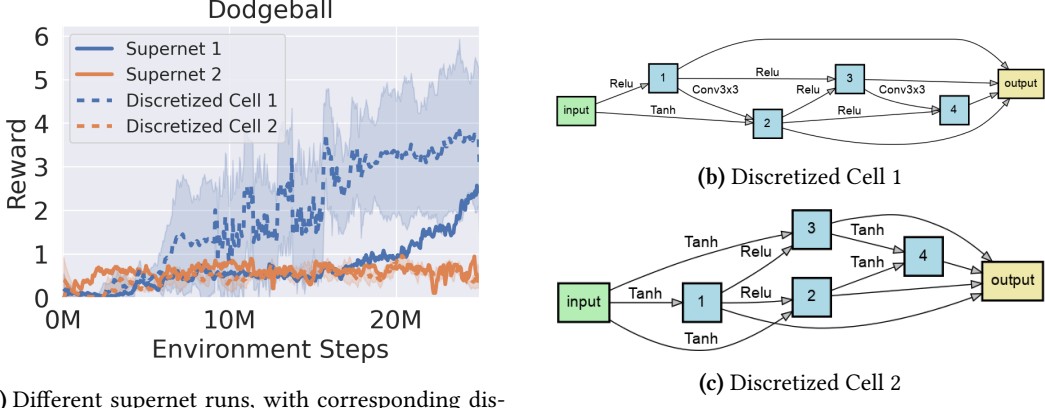

(a) Different supernet runs, with corresponding discretized cell (depths $64 \times 5$) training curves.

**Figure 11:** **(a)** Two different supernets trained on the Dodgeball environment using Rainbow, with corresponding discretized cells evaluated using 3 random seeds. **(b)** Discretized cell from Supernet 1. Note the similarity to regular Conv3x3 + ReLU designs. **(c)** Discretized cell from Supernet 2, which uses too many Tanh nonlinearities, known to cause vanishing gradient effects.

We investigate further in Appendix D.3, Figure 19, and find that supernet and discrete cell rewards are indeed correlated, after adjusting for environment-dependent factors, suggesting that search quality can be improved via both better supernet training (Raileanu et al., 2020; Zhang et al., 2021), as well as better discretization procedures (Liang et al., 2019; Wang et al., 2021b).

## 4 Conclusions, Limitations, and Broader Impact Statement

**Conclusion.** Even though RL uses complex loss functions defined over nonstationary data, we empirically showed that nonetheless, DARTS is capable of improving policy architectures in a minimally invasive and efficient way (only 3x more compute time) across several different algorithms (PPO, Rainbow, SAC) and environments (Procgen, DM-Control). Our paper is the first to have comprehensively provided evidence for the applicability of softmax/gradient-based architecture search outside of standard classification and SL.

**Limitations.** To avoid confounding factors and for simplicity, our paper uses the default DARTS method. However, we outline multiple possible improvements in Appendix F, as RL is a completely new frontier for which to understand softmax routing and continuous relaxation techniques.

**Future Work and Broader Impact.** In this paper, we have provided concrete evidence that architecture search can be conducted practically and efficiently in RL, via DARTS. We believe that this could start important initiatives into finding better and more efficient (Cai et al., 2019) architectures for large-scale robotics (James et al., 2019; Akinola et al., 2021), transferable architectures in offline RL (Levine et al., 2020), as well as RNNs for memory (Pritzel et al., 2017; Fortunato et al., 2019; Kapturowski et al., 2019) and adaptation (Duan et al., 2016; Wang et al., 2017). Other NAS methods' applicability in RL may also be investigated, especially ones which utilize blackbox optimization controllers, such as multi-trial evolution (Real et al., 2019) or ENAS (Pham et al., 2018).

Furthermore, we believe our work's potential negative impacts are generally equivalent to ones found in general NAS methods, such as sacrificing model interpretability in order to achieve higher objectives. For the field of RL specifically, this may warrant more attention in AI safety when used for real world robotic pipelines. Furthermore, as with any NAS research, the initial phase of discovery and experimentation may contribute to carbon emissions due to the computational costs of extensive tuning. However, this is usually a means to an end, such as an efficient search algorithm, which this paper proposes with no extra hardware costs.

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
