# Appendix

## A Efficiency Metrics

We provide computational efficiency metrics in Table 4, where we find that the practical wall-clock time required for training the supernet (i.e. the search cost) is very comparable with DARTS in SL (Liu et al., 2019), requiring only a few GPU days. We do note that unlike SL where the vast majority of the cost is due to the network, RL time cost is partially based on non-network factors such as environment simulation, and thus wall-clock times may change depending on specific implementation.

| Network | Training Cost in GPU Days (w/ specific algorithm) |
|---|---|
| IMPALA-CNN | 1 (PPO), 0.5 (Rainbow) |
| "Classic" Supernet | 2.5 (PPO) |
| "Micro" Supernet | 1.5 (Rainbow) |
| CIFAR-10 Supernet (Liu et al., 2019) | 4 (SL/Original DARTS) |

Table 4: Computational efficiency in terms of wall-clock time, achieved on a V100 GPU. For the RL cases (PPO + Rainbow), all networks use depths of $16 \times 3$. Training cost in RL is defined as the wallclock time taken to reach 25M steps, rounded to the nearest 0.5 GPU day. We have also included reported time for DARTS in SL (Liu et al., 2019) as comparison.

## B Extended Supernet Training Results

Following Subsection 3.2, we also present a similar figure for Rainbow below, for completeness.

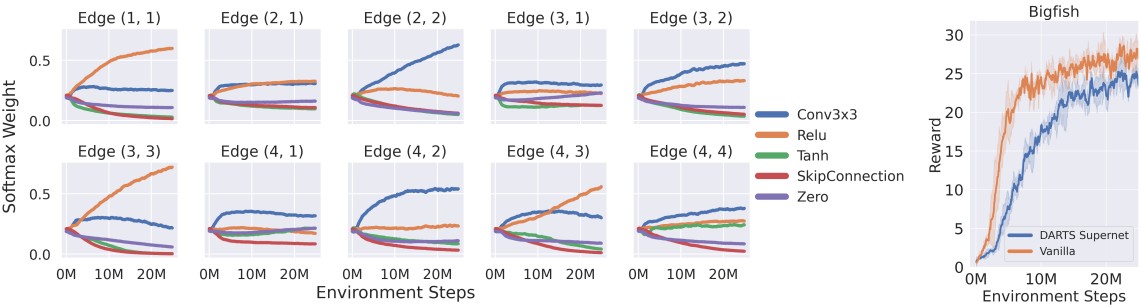

Figure 12: Analogous settings with Figure 8 using Rainbow + "Micro" search space. **Left**: Softmax weights when training Rainbow with infinite levels on Bigfish, also converging towards a sparser solution. **Right**: Sanity check for supernet when using Rainbow.

## C What Affects Supernet Training?

Given the positive training results we demonstrated in the main body of the paper, one may wonder, *can any supernet, no matter how poorly designed or setup, still train well in the RL setting?* If so, this would imply that the search method would not be producing meaningful, but instead, random signals.

We refute this hypothesis by performing ablations over our supernet training in order to have a better understanding of what components affect its performance. We ultimately show that the search space and architecture variables play a very significant role in its optimization, thus validating our method.

### C.1 Role of Search Space

We remove the ReLU nonlinearities from the "Classic" search space, so that $\mathcal{O}_{base}$ = {Conv3x3, Conv5x5, Dilated3x3, Dilated5x5} and thus the DARTS cell consists of only linear operations. As

shown in Fig. 13, this leads to a dramatic decrease in supernet performance, providing evidence that the search space matters greatly.

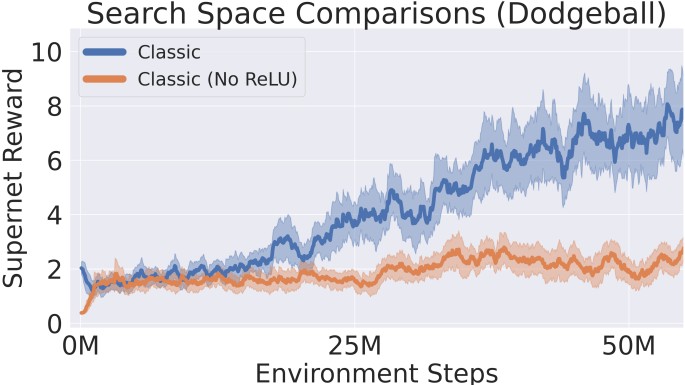

**Figure 13**: Supernet training using PPO on Dodgeball with infinite levels, when using the "Classic" search space with/without ReLU nonlinearities, under the same hyperparameters.

## C.2 Uniform Architecture Variables

We further demonstrate the importance of the architecture variables $\alpha$ on training. We see that in Fig. 14, freezing $\alpha$ to be uniform throughout training makes the Rainbow agent unable to train at all. This suggests that it is crucial for $\alpha$ to properly route useful operations throughout the network.

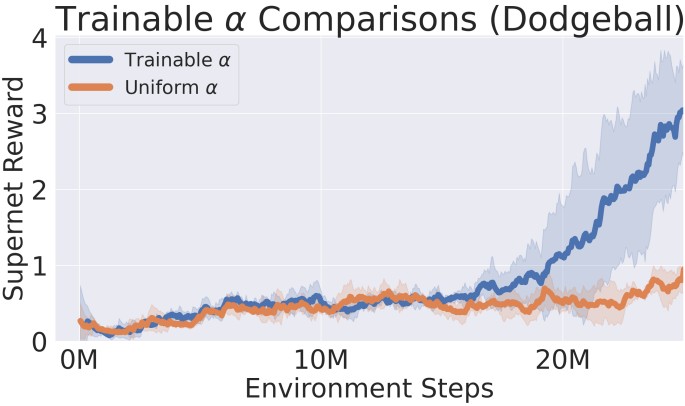

**Figure 14**: Supernet training using Rainbow on Dodgeball with infinite levels, when using the "Micro" search space with/without trainable architecture variables $\alpha$, under the same hyperparameters.

## D  What Affects Discrete Cell Performance?

### D.1  Softmax Weights vs Discretization

As seen from Figure 8 in the main body of the paper, the DARTS supernet strongly downweights Conv5x5+ReLU operations when using the "Classic" search space with PPO. In order to verify the predictive power of the softmax weights, as a proxy, we thus also performed evaluations when using purely 3x3 or 5x5 convolutions on a large IMPALA-CNN with $64 \times 5$ depths. We see that the Conv3x3 setting indeed outperforms Conv5x5, corroborating the results in which during training, $\alpha$ strongly upweights the Conv3x3+ReLU operation and downweights Conv5x5+ReLU.

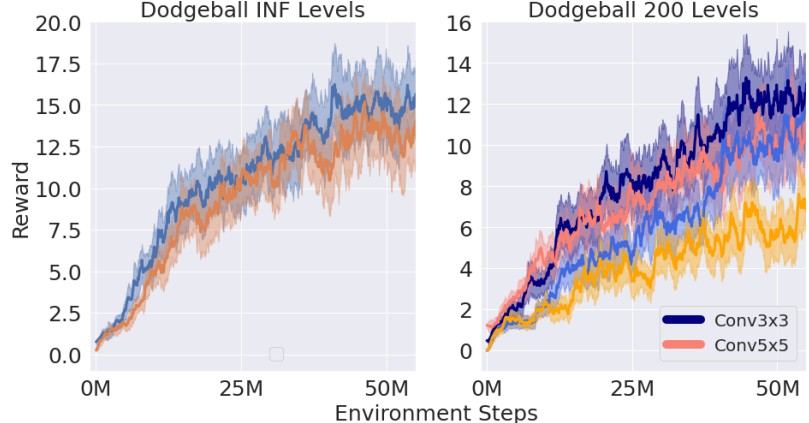

**Figure 15**: Large IMPALA-CNNs evaluated on Dodgeball using either Infinite or 200 levels with PPO. For the 200 level setting, lighter colors correspond to test performance.

### D.2  Discrete Cell Evolutions

Along with Figure 10 in the main body of the paper, we also compare extra examples of discretizations before and after supernet training, to display reasonable behaviors induced by the trajectory of $\alpha$.

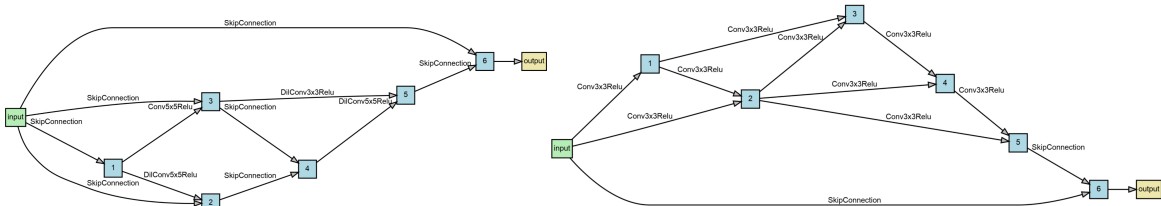

**Figure 16**: Comparison of discretized cells before and after supernet training, on Starpilot using PPO with $I = 6$ nodes.

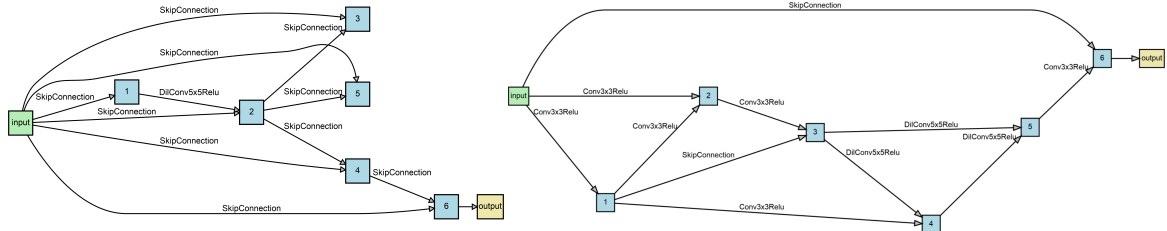

**Figure 17**: Comparison of discretized cells before and after supernet training, on Plunder using PPO with $I = 6$ nodes.

In Figure 16, we use PPO with the "Classic" search space, but instead use $(N, R, I) = (1, 0, 6)$ along with outputting the last node (instead of concatenation with a Conv1x1 for the output) to

allow a larger normal cell search space and graph topology. In Figure 17, the discretized cell initially uses a large number of skip connections as well as dead-end nodes. However, at convergence, it eventually utilizes all nodes to compute the final output. Curiously, we find that the skip connection between the input and output appears commonly throughout many searches.

For the Rainbow setting, in Figure 9 in the main body of the paper, we saw that when the search process is successful, the supernet's training trajectory induces discretized cells which improve evaluation performance as well. The cells discovered later generally perform better than cells discovered earlier in the supernet training process. In Figure 18, we show more examples of such evaluation curves.

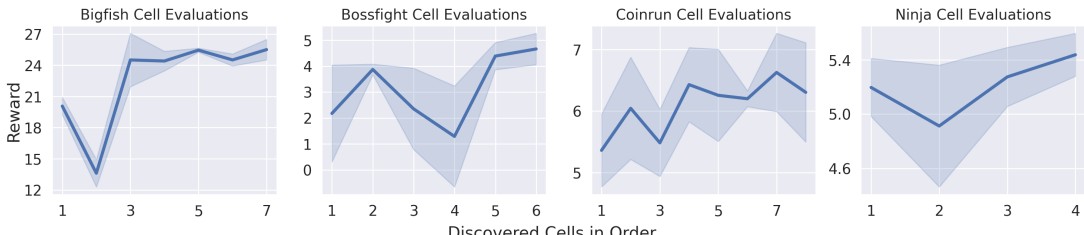

**Figure 18**: Evaluated discretized cells discovered throughout training the supernet with Rainbow. To save computation, we evaluate every 2nd cell that was discovered.

## D.3 Correlation Between Supernet and Discretized Cells

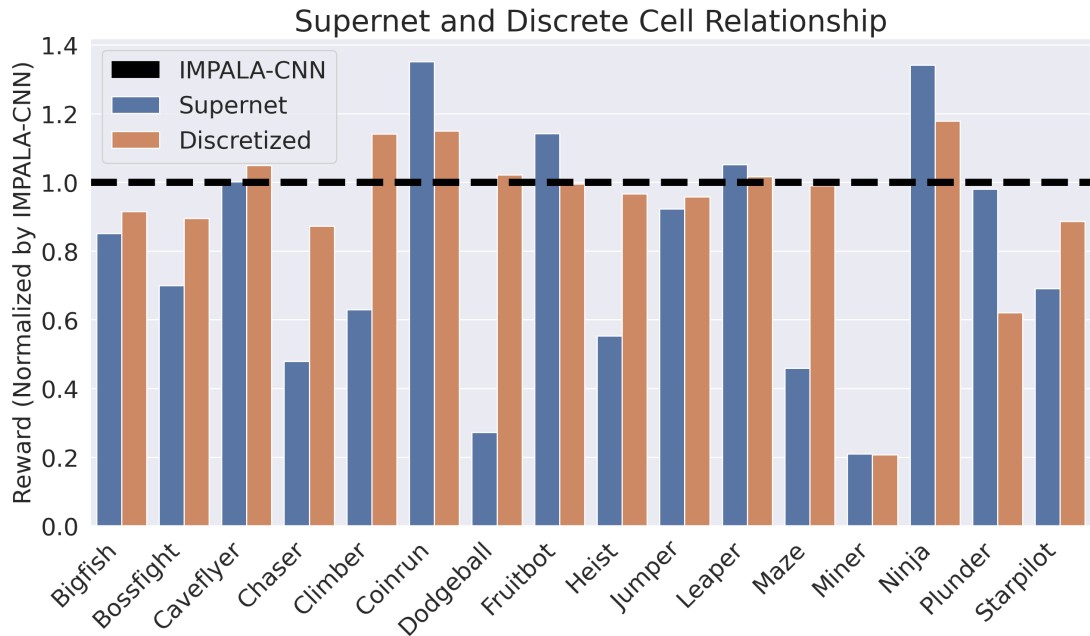

**Figure 19**: Supernet and their corresponding discrete cell rewards across all environments in Procgen using Rainbow, after normalizing using IMPALA-CNN's performances. Thus, the black dashed line at 1.0 corresponds to IMPALA-CNN.

Given that the discretized cell only explicitly depends on architecture variables $\alpha$ and not necessarily model weights $\theta$, one may wonder: *Is there a relationship between the rewards of the supernet and of its corresponding discretized cell?* For instance, the degenerate/underperformance setting mentioned in Section 3.3 and Appendix D can be thought of as an extreme scenario. At the same time, there could be an integrality gap, where there the discretization process $\delta(\alpha)$ produces cells which give different rewards than the supernet.

In order to make such an analysis comparing rewards, we first must prevent confounding factors arising from Rainbow's natural performance on an environment regardless of architecture. We thus first divide the supernet and discrete cell scores by the score obtained by the IMPALA-CNN baseline, where the baseline and discrete cells all used depths of $64 \times 3$.

In Figure 19, when using Rainbow and observing across environments, we find both high correlation and also integrality gaps: for some environments such as Ninja and Coinrun, there is a significant correlation between supernet and discrete cell rewards, while for other environments such as Dodgeball, there is a significant gap. This suggests that search quality can be improved via both better supernet training such as using hyperparameter tuning or data augmentation (Raileanu et al., 2020), as well as better discretization procedures such as early stopping and stronger pruning (Chu et al., 2020a; Liang et al., 2019; Wang et al., 2021b).

## E  Numerical Scores

In Tables 5 and 5b, we display the average normalized reward after 25M steps, as standard in Procgen (Cobbe et al., 2020), for a subset of environments in which RL-DARTS performs competitively. The normalized reward for each environment is computed as $R_{norm} = (R - R_{min})/(R_{max} - R_{min})$ where $R_{max}$ and $R_{min}$ are calculated using a combination of theoretical maximums and PPO-trained agents, and can be found in (Cobbe et al., 2020).

| Env | IMPALA-CNN Baseline | RL-DARTS (Discrete) | Random Search |
|---|---|---|---|
| Bigfish | **0.60** | **0.60** | 0.42 |
| Bossfight | 0.75 | 0.73 | **0.81** |
| Caveflyer | 0.75 | 0.47 | **0.85** |
| Chaser | **0.71** | 0.55 | 0.15 |
| Climber | 0.69 | **0.90** | 0.61 |
| Coinrun | **0.91** | 0.53 | 0.8 |
| Dodgeball | 0.53 | **0.59** | 0.29 |
| Fruitbot | **0.92** | **0.93** | 0.83 |
| Heist | 0.72 | **0.89** | 0.38 |
| Jumper | 0.62 | 0.76 | **1.0** |
| Leaper | 0.2 | **0.28** | -0.28 |
| Maze | **1.0** | **1.0** | 0.0 |
| Miner | 0.74 | **0.85** | 0.70 |
| Ninja | **0.87** | 0.69 | 0.38 |
| Plunder | 0.57 | **0.76** | 0.43 |
| Starpilot | 0.71 | **0.73** | 0.40 |

**(a)** PPO + Classic

| Env | IMPALA-CNN Baseline | RL-DARTS (Discrete) | Random Search |
|---|---|---|---|
| Bigfish | **0.71** | 0.65 | 0.60 |
| Bossfight | **0.54** | **0.48** | 0.45 |
| Caveflyer | -0.05 | **-0.03** | **-0.01** |
| Chaser | **0.30** | **0.26** | 0.22 |
| Climber | **-0.04** | **-0.02** | **-0.05** |
| Coinrun | 0.06 | **0.21** | 0.15 |
| Dodgeball | **0.57** | **0.59** | -0.05 |
| Fruitbot | **0.68** | **0.68** | **0.70** |
| Heist | **-0.48** | **-0.49** | **-0.47** |
| Jumper | **0.21** | 0.18 | 0.17 |
| Leaper | -0.07 | -0.07 | **-0.03** |
| Maze | **0.76** | **0.74** | 0.51 |
| Miner | **0.35** | -0.03 | 0.11 |
| Ninja | 0.03 | 0.13 | **0.28** |
| Plunder | **0.14** | 0.02 | 0.03 |
| Starpilot | **0.91** | 0.80 | 0.76 |

**(b)** Rainbow + Micro

**Table 5**: Normalized Rewards in ProcGen across different search methods, evaluated at 25M steps with depths $64 \times 3$. Largest scores on the specific environment (as well as values within 0.03 of the largest) are **bolded**.

# F Possible Improvements for Future Work

These include:

1. Discretization Changes: One may consider discretization based on the total reward $J(\pi_{\theta,\alpha})$, which may provide a better signal for the correct discrete architecture. This is due to the fact that the relative strengths of operation weights from $\alpha$ may not correspond to the best choices during discretization. (Wang et al., 2021b) considers iteratively pruning edges from the supernet based on maximizing validation accuracy changes. For RL, this would imply a variant of discretization dependent on multiple calculations of $J(\pi_{\theta^*,\delta_1(\alpha^*)}) - J(\pi_{\theta^*,\delta_2(\alpha^*)})$ where $\theta^*$ consists the weights obtained during supernet training, as well as fine-tuning $J(\pi_{\theta^*,\delta_1(\alpha^*)})$ at every pruning step. These changes, in addition to the inherently noisy evaluations of $J(\cdot)$, greatly increase the complexity of the discretization procedure, but are worth exploring in future work.

2. Changing the Loss / Regularization: Throughout this paper, we have found that vanilla DARTS is able to train by simply optimizing $\alpha$ with respect to the loss, even though in principle, the loss is not strongly correlated to the actual reward in RL. Thus, it is curious to understand whether loss-based metrics or modifications may help improve RL-DARTS. One such modification is based on the observation that certain RL losses may not be required for training $\alpha$. In PPO, the entropy loss of $\pi_\theta$ may not be necessary or useful for improving the search quality of $\alpha$, and thus it may be better to perform a two step update by providing a different loss for $\alpha$. One may also consider searching for two separate encoders via two supernets, since both PPO and Rainbow feature separate networks, e.g. the policy $\pi_{\theta_1,\alpha_1}$ and value function $V_{\theta_2,\alpha_2}$ for PPO and advantage function $A_{\theta_1,\alpha_1}$ and value function $V_{\theta_2,\alpha_2}$ for Rainbow.

3. Signaling Metrics and Early Stopping: Observing metrics throughout training allows for early stopping, which can reduce search cost and provide better discrete cells. This includes metrics such as the strength of certain operation weights (Liang et al., 2019) as well as the Hessian with respect to $\alpha$ throughout training, i.e. $\nabla^2_\alpha \mathcal{L}(\theta, \alpha)$ as found in (Zela et al., 2020). Furthermore, inspired by performance prediction methods (Mellor et al., 2020; Luo et al., 2018), one may analyze metrics such as the Jacobian Covariance, via the score defined to be the $-\sum_{i=1}^{B} \left[ \log(\sigma_i + \varepsilon) + (\sigma_i + \varepsilon)^{-1} \right]$ where $\varepsilon = 10^{-5}$ is a stability constant and $\sigma_1 \leq \ldots \leq \sigma_B$ are the eigenvalues of the correlation matrix corresponding to the Jacobian $J = \left[ \frac{\partial f}{\partial s_1}, \ldots, \frac{\partial f}{\partial s_B} \right]^T$ with $B$ input images $\{s_1, \ldots, s_B\}$.

   This metric/predictor has been found to be a strong signal for accuracy in SL NAS among many previous predictors (White et al., 2021). However, for the RL case, just like the loss, the mentioned metrics must be defined with respect to the current replay buffer $\mathcal{D}$, and thus raises the question of what type of data is to be used for calculating these metrics. When using a reasonable variant where the data is collected from a pretrained policy, we found that methods such as Jacobian Covariance did not provide meaningful feedback.

4. Supernet Training: As seen from the results in Figure 11 (main body) and Appendix D.3, there is a correlation between supernet and discrete cell performances. However, this is affected by the environment used as well as integrality gaps between the continuous relaxation and discrete counterparts, and thus further exploration is needed before concluding that improving the supernet training leads to better discrete cell performances. In any case, reasonable methods of improving the supernet can involve DARTS-agnostic modifications to the RL pipeline, including data augmentation (Kostrikov et al., 2020; Raileanu et al., 2020) as well as online hyperparameter tuning (Parker-Holder et al., 2020, 2021b). Simple hyperparameter tuning (e.g. on the softmax temperature for calculating $p_o^{(i,j)}$'s) also can be effective.

## G Hyperparameters

In our code, we entirely use Tensorflow 2 for auto-differentation, as well as the April 2020 version of Procgen. For compute, we either used P100 or V100 GPUs based on convenience and availability. Below are the hyperparameter settings for specific methods. For all training curves, we use the common standard for reporting in RL (i.e. plotting mean and standard deviation across 3 seeds).

### G.1 DARTS

Initially, we swept the softmax temperature in order to find a stable default value that could be used for all environments. For PPO, the sweep was across the set $\{5.0, 10.0, 15.0\}$. For Rainbow, the sweep was across $\{10.0, 20.0, 50.0\}$.

For tabular reported scores in Figures 5 and 5b, we used a consistent softmax temperature of 5.0 for PPO, and 10.0 for Rainbow.

### G.2 Rainbow-DQN

We use Acme (Hoffman et al., 2020) for our code infrastructure. We use a learning rate $5 \times 10^{-5}$, batch size 256, n-step size 7, discount factor 0.99. For the priority replay buffer (Schaul et al., 2016), we use priority exponent 0.8, importance sampling exponent 0.2, replay buffer capacity 500K. For particular environments (Bigfish, Bossfight, Chaser, Dodgeball, Miner, Plunder, Starpilot), we use n-step size 2 and replay buffer capacity 10K. For C51 (Bellemare et al., 2017), we use 51 atoms, with $v_{min} = 0, v_{max} = 1.0$. As a preprocessing step, we normalize the environment rewards by dividing the raw rewards by the max possible rewards reported in (Cobbe et al., 2020).

### G.3 PPO

We use TF-Agents (Guadarrama et al., 2018) for our code infrastructure, along with equivalent PPO hyperparameters found from (Cobbe et al., 2020). Due to necessary changes in minibatch size when applying DARTS modules or networks with higher GPU memory usage, we thus swept learning rate across $\{1 \times 10^{-4}, 2.5 \times 10^{-4}, 5 \times 10^{-4}\}$ and number of epochs across $\{1, 2, 3\}$.

For all models, we use a maximum power of 2 minibatch size before encountering GPU out-of-memory issues on a standard 16 GB GPU. Thus, for a $16 \times 3 = [16, 16, 16]$ DARTS supernet, we set the minibatch size to be 256, which is also used for evaluation with a $64 \times 3 = [64, 64, 64]$ discretized CNN. Our hyperparameter gridsearch for the evaluation led to an optimal setting of learning rate $= 1 \times 10^{-4}$ and number of epochs $= 1$.

### G.4 SAC

We use open-source code found in https://github.com/google-research/pisac, although we disabled the predictive information loss to use only regular SAC. The baseline architecture is a 4-layer convolutional architecture found in https://github.com/google-research/pisac/blob/master/pisac/encoders.py. Image-based observations are resized to $64 \times 64$ with a frame-stacking of 3. Both our DARTS supernet and discrete cells use $N = 3, I = 4, K = 1$ using the "Micro" search space, with convolutional depths of 32 to remain fair to the baseline.

## G.5 Training Procedure

Below are PPO (Schulman et al., 2017) and Rainbow-DQN (Hessel et al., 2018) RL-DARTS variants, which provide an example of the specific training procedure we use. Exact loss definitions and data collection procedures can be found in their respective papers.

---
**Algorithm 2**: RL-DARTS with PPO
---

Supernet training:

    Setup supernet encoder $f_{\theta_e,\alpha}$ with weights $\theta_e$.

    Initialize policy and value head projection weights $W_\pi \in \mathbb{R}^{d,|\mathcal{A}|}, W_v \in \mathbb{R}^{d,1}$.

    Collect all trainable weights $\theta = \{\theta_e, W_\pi, W_v\}$.

    Setup policy $\pi_{\theta,\alpha}(s) \sim \text{softmax}(W_\pi \cdot f_{\theta_e,\alpha}(s))$.

    Setup value function $V_{\theta,\alpha}(s) = W_v \cdot f_{\theta_e,\alpha}(s)$.

    Define standard PPO loss $\mathcal{L}(\theta, \alpha)$ using $\pi_{\theta,\alpha}$ and $V_{\theta,\alpha}$.

    Perform PPO training via collecting data from $\pi_{\theta,\alpha}$ and SGD with $\nabla_{\theta,\alpha}\mathcal{L}(\theta, \alpha)$.

    Collect $\alpha^*$ from previous training procedure.

Discretization:

    Let $\delta(\alpha^*)$ be the discrete cell constructed via Algorithm 4.

Evaluation:

    Setup discretized cell encoder $f_{\phi_e,\delta(\alpha^*)}$.

    Initialize policy and value head projection weights $W'_\pi \in \mathbb{R}^{d,|\mathcal{A}|}, W'_v \in \mathbb{R}^{d,1}$.

    Collect all trainable weights $\phi = \{\phi_e, W'_\pi, W'_v\}$.

    Setup policy $\pi_{\phi,\delta(\alpha^*)}(s) \sim \text{softmax}(W'_\pi \cdot f_{\phi_e,\delta(\alpha^*)}(s))$.

    Setup value function $V_{\phi,\delta(\alpha^*)}(s) = W'_v \cdot f_{\phi_e,\delta(\alpha^*)}(s)$.

    Define standard PPO loss $\mathcal{L}_{\delta(\alpha^*)}(\phi)$ using $\pi_{\phi,\delta(\alpha^*)}$ and $V_{\phi,\delta(\alpha^*)}$.

    Perform PPO training via collecting data from $\pi_{\phi,\delta(\alpha^*)}$ and SGD with $\nabla_\phi \mathcal{L}_{\delta(\alpha^*)}(\phi)$.

    Report final policy reward.

---

---
**Algorithm 3**: RL-DARTS with Rainbow. Note that we do not use noisy nets in this implementation.
---

Supernet training:

    Setup supernet encoder $f_{\theta_e,\alpha}$ with weights $\theta_e$.

    Initialize dueling network projections $W_v \in \mathbb{R}^{d,1}, W_a \in \mathbb{R}^{d,|\mathcal{A}|}$.

    Collect all trainable weights $\theta = \{\theta_e, W_v, W_a\}$.

    Setup value network $V_{\theta,\alpha}(s) = W_v \cdot f_{\theta_e,\alpha}(s)$.

    Setup advantage network $A_{\theta,\alpha}(s, a) = W_a \cdot f_{\theta_e,\alpha}(s)$.

    Setup Q-network $Q_{\theta,\alpha}(s, a) = V_{\theta,\alpha}(s) + A_{\theta,\alpha}(s, a) - \frac{1}{|\mathcal{A}|}\sum_{a'\in\mathcal{A}} A_{\theta,\alpha}(s, a')$.

    Define standard Rainbow loss $\mathcal{L}(\theta, \alpha)$ using $Q_{\theta,\alpha}$.

    Perform Rainbow training via collecting data from $Q_{\theta,\alpha}$ and SGD with $\nabla_{\theta,\alpha}\mathcal{L}(\theta, \alpha)$.

    Collect $\alpha^*$ from previous training procedure.

Discretization

    Let $\delta(\alpha^*)$ be the discrete cell constructed via Algorithm 4.

Evaluation

    Setup discretized cell encoder $f_{\phi_e,\delta(\alpha^*)}$ with weights $\phi_e$.

    Initialize dueling network projections $W'_v \in \mathbb{R}^{d,1}, W'_a \in \mathbb{R}^{d,|\mathcal{A}|}$

    Collect all trainable weights $\phi = \{\phi_e, W'_v, W'_a\}$

    Setup value network $V_{\phi,\delta(\alpha^*)}(s) = W'_v \cdot f_{\phi_e,\delta(\alpha^*)}(s)$

    Setup advantage network $A_{\phi,\delta(\alpha^*)}(s, a) = W'_a \cdot f_{\phi_e,\delta(\alpha^*)}(s)$

    Setup Q-network $Q_{\phi,\delta(\alpha^*)}(s, a) = V_{\phi,\delta(\alpha^*)}(s) + A_{\phi,\delta(\alpha^*)}(s, a) - \frac{1}{|\mathcal{A}|}\sum_{a'\in\mathcal{A}} A_{\phi,\delta(\alpha^*)}(s, a')$

    Define standard Rainbow loss $\mathcal{L}_{\delta(\alpha^*)}(\phi)$ using $Q_{\phi,\delta(\alpha^*)}$.

    Perform Rainbow training via collecting data from $Q_{\phi,\delta(\alpha^*)}$ and SGD with $\nabla_\phi \mathcal{L}_{\delta(\alpha^*)}(\phi)$.

    Report final policy reward.

---

---

**Algorithm 4**: Discretization Procedure.

---

Argmax:

For $(i, j)$ across all edges:

Define edge strength $w_{i,j} = \max_{o \in \mathcal{O}, o \neq zero} p_o^{(i,j)}$.

Define edge op $o_{(i,j)} = \arg\max_{o \in \mathcal{O}, o \neq zero} p_o^{(i,j)}$.

Prune:

For node $j$ in all intermediate nodes:

Sort input edge weights $w_{i_1,j} \geq w_{i_2,j} \geq \ldots$

Retain only top $K$ edges $(i_1, j), \ldots, (i_K, j)$ and corresponding ops $o_{(i_1,j)}, \ldots, o_{(i_K,j)}$
in final cell.

---

Note that both RL-DARTS procedures can also be summarized in terms of raw code as simple one-line edits to the image encoder used (compressing the rest of the regular RL training pipeline code):

```
def train(feature_encoder):
    """Initial RL algorithm setup"""
    ...
    extra_variables = Wrap(feature_encoder)
    all_trainable_variables =
        [feature_encoder.trainable_variables(), extra_variables]
    """Rest of RL algorithm setup"""
    ...
    apply_gradients(loss, all_trainable_variables)
```

Thus, the 3-step RL-DARTS procedure from Section 2 can be seen as:

```
DARTSSuperNet = MakeSuperNet(ops, num_nodes) # Setup
train(DARTSSuperNet) # Supernet training
DiscretizedNet = DARTSSuperNet.discretize() # Discretization
train(DiscretizedNet) # Evaluation
```

## H Miscellaneous

### H.1 Search Space Size

Let $O_{nz} = |\mathcal{O}| - 1$, the number of non-zero ops in $\mathcal{O}$. For a cell (normal or reduction), the first intermediate node can only be connected to the input via a single op, and thus the choice is only $O_{nz}$. However, later intermediate nodes use $K = 2$ inputs which leads to a choice size of $O_{nz}^K \times \binom{i}{K}$ where $i$ is the index of the intermediate node. Thus the total number of possible discrete cells is $O_{nz} \cdot \prod_{i=2}^{I} \left( O_{nz}^K \times \binom{i}{K} \right)$.

For the "Classic" search space, there are both normal and reduction cells to be optimized, with number of non-zero normal ops $O_{nz,N} = 5$ and number of non-zero reduction ops $O_{nz,R} = 4$, with $I = 4, K = 2$ for both. This leads to a total configuration size of $\left[ O_{nz,N} \cdot \prod_{i=2}^{4} \left( O_{nz,N}^2 \times \binom{i}{2} \right) \right] \times \left[ O_{nz,R} \cdot \prod_{i=2}^{4} \left( O_{nz,R}^2 \times \binom{i}{2} \right) \right] \approx 4 \times 10^{11}$.

For the "Micro" search space, since we do not use reduction cells in order to simplify visualizations and ablation studies, $O_{nz,N} = 4$ with $I = 4, K = 2$. This gives $\left[ O_{nz,N} \cdot \prod_{i=2}^{4} \left( O_{nz}^2 \times \binom{i}{2} \right) \right] \approx 3 \times 10^5$, which is comparable to the search space of size $5^6 \approx 1.5 \times 10^4$ in NASBENCH-201 (Dong and Yang, 2020).