# OpenReview forum: "Differentiable Architecture Search for Reinforcement Learning"
_automl.cc/AutoML/2022/Track/Main — AutoML-Conf 2022 (Main Track)_

### Official Review · Reviewer_saSv · 2022-03-31

**Potential Impact On The Field Of Automl Rating:** 4
**Technical Quality And Correctness Rating:** 3
**Clarity Rating:** 4

**Summary Of Contributions:**

This paper explores the use of Neural Architecture Search, more specifically the popular DARTS algorithm, for reinforcement learning.
More precisely, this paper conducts a thorough analysis of the performance and the behavior of the DARTS algorithm at the cell level when training in the standard ProcGen environment.

**Clarity:**

This paper is well presented and coherent; the language is clear and easily understandable.

**Overall Review:**

This paper is well written and tackles the interesting question of using NAS approaches (DARTS in particular) for RL.
The analysis is well conducted and each claim is supported by multiple experiments and empirical results.
However, some results are still quite weak in the sense that the performance of the DARTS algorithm does not appear significantly better than standard RL approaches on most environments while the algorithm is reported to need 3 times more computation time.

This paper claims to be the first one to try to apply DARTS in the RL setting, and although I'm not specifically familiar with the DARTS literature, I did not find any counter-examples.
Moreover, the paper only focuses on the basic DARTS algorithm and not on all improvements proposed since, so I think the results are easily good enough to show that NAS approaches are relevant for RL.

**Potential Impact On The Field Of Automl:**

In my opinion, this paper is important for the field as it extends the use of NAS methods to the RL domain and shows that these approaches are general enough to be applied in various contexts.
Although the presented algorithm is not new, the analysis conducted in this work is interesting and will benefit both the NAS community and the RL community.

**Reproducibility:**

The reproducibility checklist is included in the paper, and the items are reasonable.
According to the authors, the code will be released on a public Github (hidden for anonymity reasons for now).

**Review Confidence:**

3: You are fairly confident in your assessment. It is possible that you did not understand some parts of the submission or that you are unfamiliar with some pieces of related work.

**Review Rating:**

5: Accept, good paper

**Review Summary:**

This paper seems to be the first one to apply DARTS to RL and to show that this is an interesting approach.
The results are not really better than standard RL approaches, but the discussion is well conducted and this work acts more as a proof of concept.
In consequence I would recommend to accept this paper.

**Technical Quality And Correctness:**

In order to evaluate the DARTS algorithm, this paper proposes experiments that are well justified and correctly conducted.
Many details are taken into account to ensure the fairness of the different comparisons to other approaches.
The empirical analysis is interesting and based on strong observations, and although not necessary, the appendices give plenty of details further supporting the conclusions of the article.

---

### Official Review · Reviewer_pzqS · 2022-04-02

**Potential Impact On The Field Of Automl Rating:** 3
**Technical Quality And Correctness Rating:** 3
**Clarity Rating:** 3

**Summary Of Contributions:**

The paper studies whether and to what extent the Differentiable Architecture Search (DARTS) is applicable to Reinforcement Learning (RL). They build a network called RL-DARTS, which applies the original DARTS to search architectures for RL algorithms such as PPO, Reinbow, SAC. For the experiments, they use Procgen environment for discrete action spaces and DM-control benchmark for continuous action space study. For Procgen benchmark, they performed both multi-game search and single-game search with PPO. In the multi-game search, they train and supernet across 16 games and achieved up to 250% performance over the IMPALA-CNN baseline, and in the single-game search, their algorithm can find game-specific architectures from scratch which outperforms IMPALA-CNN and random search. They have also performed experiments on training the supernet end-to-end with minimal hyperparameter tuning, which costs only 3 times more compute time to obtain extensive efficiency. Finally, they demonstrate that the discrete cell can also be improved on both quantitative and quantitative aspects with the default discretization, and they claim that the supernet and discrete cell rewards are correlated according to their experiments. In conclusion, they claim that jointly optimize both \theta and \alpha can provide concrete evidence that DARTS can be conducted efficiently in RL.


**Clarity:**

I do not have much experience with DARTS and NAS, so I don't have some good advice for future improvements. If the authors could give some more related works explained would be more helpful.

**Overall Review:**

Positive:
- Well-written and well structured.
- The Methodology is well explained.
- The experiments are well conducted to answer the authors' research questions.

Negative:
- A minor miss explanation is that SL states Supervised Learning, which might cause some confusion to people who are not familiar with it.
- The comparison of this work with the previous NAS for RL studies is limited and not very clear to me.



**Potential Impact On The Field Of Automl:**

In my opinion, this study and their experimental results can somewhat find better architectures for RL algorithms, and performs much better than random search. However, in comparison to the IMPALA-CNN baseline, the improvement is not significant.

**Reproducibility:**

The environmental settings, training procedures, and the code are all provided in the appendix, so I believe they are reproduceable.

**Review Confidence:**

3: You are fairly confident in your assessment. It is possible that you did not understand some parts of the submission or that you are unfamiliar with some pieces of related work.

**Review Rating:**

4: Marginally above the acceptance threshold (use sparsely)

**Review Summary:**

In general, I think it is a well-written paper with comprehensive comparison experiments. According to their results, DARTS can indeed be applied to RL with some extra (3 times) computing time to obtain better performance. Although, in my opinion, the novelty and the performance improvement of the idea is not very significant, I think it is acceptable.

**Technical Quality And Correctness:**

Their experimental results are clear and sound reasonable to me, the ideas and conclusions are also clearly presented.

---

### Official Review · Reviewer_8kUq · 2022-04-04

**Potential Impact On The Field Of Automl Rating:** 3
**Technical Quality And Correctness Rating:** 3
**Clarity:** 1. Maybe I've missed some details her…
**Clarity Rating:** 3

**Summary Of Contributions:**

This paper investigates how to apply DARTS, a differentiable neural archirecture search method that previously has only been used in supervised learning, to reinforcement learning.
The authors argue that this is not a trivial application of DARTS to a new domain, and propose several key challenges due to the different optimization paradigm of RL compared to SL, including (1) nonstationary data collection process dependent on the policy itself, and (2) mismatch between the RL objective (also the objective for DARTS update) and the true goal of maximizing return in RL.
The authors propose a slightly modified version of DARTS, which jointly optimizes both the model parameters and supernet weights. They compare to different baseline methods on different benchmarks, which illustrates that DARTS can work in RL domain.

**Overall Review:**

Pos:
1. This paper provides a first investigation on how DARTS applies to NAS to RL, which could be valuable for future work in (differentiable) NAS for RL.
2. The paper includes detailed experiments to check different aspects of DARTS during the learning process, providing both quantitative and qualitative results which proves that DARTS indeed learns something meaningful in RL domain.

Neg:
1. As mentioned in the "Technical quality" part, I think the key challenges of applying DARTS to RL as mentioned by the authors in Section 3 is not well studied in the experimental section. I think a deeper look at these could significantly improves the quality of this paper, instead of just proving a fact that DARTS can work in RL domain.
2. I don't see a clear advantage of RL-DARTS compared to the baselines w.r.t. the average performance, and the performance gap variance seems to be large across seeds and different tasks. It will be better if the paper can check the stability of RL-DARTS learning process and provide some insights on why this happens and how to improve it.

**Potential Impact On The Field Of Automl:**

With abundant experiments, this paper proves that DARTS can apply to RL domain with large-scale policy model, which could serve as a good benchmark and baseline for future work that want to investigate differentiable NAS in RL domain.

**Reproducibility:**

Haven't checked the code yet, but the checklist looks ok to me.

**Review Confidence:**

3: You are fairly confident in your assessment. It is possible that you did not understand some parts of the submission or that you are unfamiliar with some pieces of related work.

**Review Rating:**

4: Marginally above the acceptance threshold (use sparsely)

**Review Summary:**

This paper takes a first step of applying DARTS to RL with large-scale model size, which is a good contribution to NAS in RL. The methodology novelty is limited, but the experimental evaluation is convincing for the fact that DARTS can indeed work in the RL domain.

Novelty and contribution of the paper can be further improved if more insights can be provided on why DARTS is easy / hard to work on RL supported by carefully designed experiments.

**Technical Quality And Correctness:**

Genrally I think the technical quality of this paper is good. The methodology novelty may be incremental compared to the original DARTS paper, but the experimental evaluation is detailed and convinces me that DARTS is indeed applicable to RL domain.

I think this paper could be of higher technical quality if it can provide more insights on why DARTS is workable / hard on RL, in addition to the fact that DARTS can indeed work on RL problems. E.g., in Section 2.1, the authors list two reasons that may hinder the application of DARTS to RL. But there seems to be no experiment that directly check these two issues. E.g., to check the nonstationary data issue, maybe some offline RL setting could be helpful here. To check the learning signal issue, maybe different objective could be compared to see how they influence the return of the NAS process.

---

### Official Review · Reviewer_x2HR · 2022-04-06

**Potential Impact On The Field Of Automl Rating:** 3
**Technical Quality And Correctness Rating:** 3
**Clarity Rating:** 3

**Summary Of Contributions:**

In this work, the authors extend the idea of Differentiable Architecture Search (DARTS) for the Reinforcement Learning domain and verify the viability of such an approach in the case of designing better neural network architecture for RL algorithms. They experimented with both discrete and continuous action space environments using the Procgen benchmark and the DeepMind Control Suite. They claim to achieve up to 250% performance improvement in some cases at the cost of 3 times greater computation time. They also performed ablation studies to answer a few insightful questions.

**Clarity:**

Overall, the paper is clear and well written. It would be great if you could add some details about the basic idea of DARTS in section 2 under DARTS Preliminaries. This will help anyone who has general knowledge about RL but not NAS.

It's better to cite some relevant papers (such as [1],[2], [3]) after the comment made in the first line of the 3rd paragraph in section 1 (at line 42).

Please note, in line 260, one should be “qualitative”.


[1] Cobbe, K., Hesse, C., Hilton, J., & Schulman, J. (2020, November). Leveraging procedural generation to benchmark reinforcement learning. In International conference on machine learning (pp. 2048-2056). PMLR.

[2] Nafi, N. M., Glasscock, C., & Hsu, W. (2021, October). Attention-based Partial Decoupling of Policy and Value for Generalization in Reinforcement Learning. In Deep RL Workshop NeurIPS 2021.

[3] Cobbe, K. W., Hilton, J., Klimov, O., & Schulman, J. (2021, July). Phasic policy gradient. In International Conference on Machine Learning (pp. 2020-2027). PMLR.

**Overall Review:**

For years, RL models have been mainly hand-designed and tailored to particular environments. This work presents an approach to learn the model architecture via gradient-based Neural Architecture Search and contributes to the general AutoRL research.

The paper clearly identified the key differences between Supervised Learning (SL) and RL. Then they attempted to answer whether these differences restrict the applicability of the DARTS to RL and answered negatively based on their experimental results. However, their novelty lies only in establishing the connection between RL and one existing DARTS method for SL by proposing some adaptations. I would like to see some effective modification of DARTS beyond the generic findings that gradient-based NAS can be applied to RL.

In Figures 3 and 4, I would like to see the plots for at least 8 environments out of 16 environments from Procgen as opposed to only 4. Otherwise, the performance improvement is not motivating enough based on the limited numeric improvement presented in Table 1 and Table 5. In line 8, it is mentioned that up to 250% performance improvement can be achieved while line 65 suggests 150% performance gain. Thus, it causes confusion about the actual performance gain. Also, comparisons with other hand-designed modifications of IMPALA-CNN architecture such as [1] may be included to show the effectiveness of the approach.

The ablation studies are well motivated and provide meaningful insights regarding the presented approach. The architectures discovered in different stages of training justify that actual learning is happening at the architecture level.

[1] Nafi, N. M., Glasscock, C., & Hsu, W. (2021, October). Attention-based Partial Decoupling of Policy and Value for Generalization in Reinforcement Learning. In Deep RL Workshop NeurIPS 2021.

**Potential Impact On The Field Of Automl:**

This paper presents the viability of gradient-based neural architecture search for the RL domain. Although the results are not decisively better in all 16 environments, this will clearly serve as a starting point for future RL researchers who want to learn architecture automatically for their algorithms.

**Reproducibility:**

They have filled out the reproducibility list and provided code for reproduction. They also provided log files to generate some plots for the Procgen environments. However, I would also like to see the necessary wrappers and associated codes for reproducing the DeepMind Control Suite results.

**Review Confidence:**

3: You are fairly confident in your assessment. It is possible that you did not understand some parts of the submission or that you are unfamiliar with some pieces of related work.

**Review Rating:**

4: Marginally above the acceptance threshold (use sparsely)

**Review Summary:**

The work deals with a crucial issue - the automated design of network architectures for RL. They empirically demonstrate that DARTS can be used to design a competitive RL architecture. This is an important finding that the community should know. Thus, I am voting for acceptance. However, there is still room for improvement in terms of the approach's efficacy across different environments.

**Technical Quality And Correctness:**

The experiments and ablations are well structured and exhaustive. Their result presents meaningful conclusions about the intrinsic issues such as supernet training, discrete cell performance progression, minimal hyperparameter training, however, the overall performance of the learned architecture cannot be well justified by the marginal improvement compared to the IMPALA-CNN baseline reported in Table 1. This also questions the generalizability of the learned architecture for each environment available in the Procgen Benchmark.

---

### Official Review · Reviewer_1pAf · 2022-04-06

**Potential Impact On The Field Of Automl Rating:** 4
**Technical Quality And Correctness Rating:** 4
**Clarity Rating:** 3

**Summary Of Contributions:**

The paper introduces the DARTS neural architecture search algorithm for the policies of reinforcement learning (RL) agents. In their experiments, the combination of the RL loss function with the updating of the architecture is explored. Second, the problems that can arise due to a possibly diverging RL agent. Finally, the discretization of the modules can lead to failure modes. The combination of DARTS and RL shows viable with several promising results.

**Clarity:**

A minor comment related to the search space definition O\_base in the DARTS preliminaries of section 2. It states that skip connections and zero operations must be contained in the DARTS search space. Is it not possible to omit these operations from the search space or is this specific to this application?

**Overall Review:**

The authors provide a large number of experiments exploring how DARTS works in the reinforcement learning framework. The experiments provide the reader with a good overview of the success and failure cases of the method. The authors identify the two important differences in the optimization of the architecture between supervised en reinforcement learning. Additionally, the supplementary material contains an extensive review of the choices for the search space design.

Moreover, the RL-DARTS method finds well-performing architectures improving over existing policy architectures and the random search baseline. Finally, it would be interesting to extend the random search baseline by comparing it with a more competitive version of random search.


**Potential Impact On The Field Of Automl:**

The paper extends neural architecture search in the field of reinforcement learning. Additionally, the report demonstrates the workings of the DARTS policy search in several experiments. This is the first paper, to the best of my knowledge, combining DARTS with an agent's policy. Moreover, the authors show potential failure cases and successful cases.

**Reproducibility:**

The reproducibility list was filled inadequately. Additionally, I was able to run the provided code for the reinforcement learning agents. For reproducibility, the ml-collections dependency was missing from the requirements.txt.

**Review Confidence:**

3: You are fairly confident in your assessment. It is possible that you did not understand some parts of the submission or that you are unfamiliar with some pieces of related work.

**Review Rating:**

5: Accept, good paper

**Review Summary:**

The existing work provides an extensive analysis of DARTS for the architecture search for reinforcement learning policies. My only recommendation would be to extend the Random Search baseline with a more competitive version.

**Technical Quality And Correctness:**

The paper introduces relevant literature and provides a large number of experiments. The experiments provide a good split between quantitative and qualitative results over a number of settings. However, it would be interesting to compare the method to a more competitive random search baseline (e.g. the work of  Li & Talwalker, 2019 on Random Search)

---

### Meta-Review · Area_Chair_tZhL · 2022-05-01

**Recommendation:** Accept
**Confidence:** 4

**Metareview:**

All reviewers agree that this paper is relevant for the AutoML community and addresses a potentially significant open research problem. The experimental evaluation convinced the reviewers that DARTS can be applied to RL (with some additional experimental results added after the reviews).

Pro:
- This paper has high significance to the AutoML community, showing how to effectively use DARTS for RL.
- The paper includes potential failure cases and a discussion of future work, which could be a good starting point for follow-up work. The experiments / method itself can be a good benchmark / baseline for future work on DARTS for RL (especially given that code was provided and will be open-sourced).
- Their implementation can match or outperform the hand-designed IMPALA-CNN baseline.
- I was happy to see that the authors adequately addressed the reviewer's feedback and updated the paper accordingly.

Room for improvement:
- More analysis on the generalisability of the learned architecture for more environments (some have been added in the rebuttal), or the efficacy of the approach across different domains
- The paper would benefit from more insights into _why_ DARTS works (or not) in RL.
- There might be more effective ways to modify DARTS and make it work for RL.

Overall I get the sense that the empirical evaluation is extensive and convincing, and that it is fair to leave the rest for future work. I therefore recommend acceptance.

---

### Decision · Program_Chairs · 2022-05-13

Accept